# Icephobic Performance of Multi-Scale Laser-Textured Aluminum Surfaces for Aeronautic Applications

**DOI:** 10.3390/nano11010135

**Published:** 2021-01-08

**Authors:** Stephan Milles, Vittorio Vercillo, Sabri Alamri, Alfredo I. Aguilar-Morales, Tim Kunze, Elmar Bonaccurso, Andrés Fabián Lasagni

**Affiliations:** 1Institute of Manufacturing Science and Engineering, Technische Universität Dresden, George-Bähr, Str. 3c, 01069 Dresden, Germany; andres_fabian.lasagni@tu-dresden.de; 2Airbus Central Research and Technology, Materials X, Willy-Messerschmitt, Str. 1, 82024 Taufkirchen, Germany; vittorio.vercillo@gmail.com (V.V.); elmar.bonaccurso@airbus.com (E.B.); 3Fraunhofer Institute for Material and Beam Technology IWS, Winterbergstr. 28, 01277 Dresden, Germany; sabri.alamri@iws.fraunhofer.de (S.A.); aguilarmoralesalfredo@gmail.com (A.I.A.-M.); tim.kunze@iws.fraunhofer.de (T.K.)

**Keywords:** multi-scale textures, aluminum, direct laser interference patterning, superhydrophobicity, icephobicity

## Abstract

Ice-building up on the leading edge of wings and other surfaces exposed to icing atmospheric conditions can negatively influence the aerodynamic performances of aircrafts. In the past, research activities focused on understanding icing phenomena and finding effective countermeasures. Efforts have been dedicated to creating coatings capable of reducing the adhesion strength of ice to a surface. Nevertheless, coatings still lack functional stability, and their application can be harmful to health and the environment. Pulsed laser surface treatments have been proven as a viable technology to induce icephobicity on metallic surfaces. However, a study aimed to find the most effective microstructures for reducing ice adhesion still needs to be carried out. This study investigates the variation of the ice adhesion strength of micro-textured aluminum surfaces treated using laser-based methods. The icephobic performance is tested in an icing wind tunnel, simulating realistic icing conditions. Finally, it is shown that optimum surface textures lead to a reduction of the ice adhesion strength from originally 57 kPa down to 6 kPa, corresponding to a relative reduction of ~90%. Consequently, these new insights will be of great importance in the development of functionalized surfaces, permitting an innovative approach to prevent the icing of aluminum components.

## 1. Introduction

Atmospheric icing includes all meteorological phenomena in which ice is accreting and accumulating on surfaces. As a consequence of icing, serious issues in different industrial sectors result, such as in telecommunications where transmission lines and towers can collapse due to the increased weight, leading to power outages [1]. The efficiency and output of renewable energy sources, such as wind and solar, can be also seriously endangered by ice formation [2]. In the aviation industry, flights can be delayed or canceled due to ice build-up on aircraft, and in the worst cases, fatal accidents can occur from icing-related events [3]. In particular, icing can take place both when aircraft are on the ground and in flight. On the ground, snow or freezing rain result in aircraft surface contamination, which can interfere with the aerodynamic properties of the aircraft or even lead to damage of the engines due to dislodged ice fragments. This requires on-ground de-icing techniques, which typically consist of hot mixtures of water and glycol, which are directly sprayed on the aircraft surfaces covered with ice or snow [4]. During the flight, icing is caused by the impingement of supercooled, micrometer-sized water droplets in clouds, which may either directly adhere upon impact or flow back as a thin film of water and collect into rivulets because of surface tension [5,6].

Most of the technical surfaces today consist of aluminum alloys, which, due to their light weight and high strength, are employed, e.g., on the leading edge of the wings [7,8,9]. The term de-icing implies the removal of ice after its formation on a surface exposed to extreme weather conditions, whereas anti-icing describes the capacity of a surface to reduce or avoid ice formation [10]. On airplanes, de-icing is achieved by the use of thermal energy [11] through hot bleed air diverted from the compressor of the engine to the location to be de-iced [10]. Alternatively, electrothermal heaters are used for areas where thermal de-icing systems are not applicable, such as propellers, spinners, nose cones, helicopter rotors, and hubs [12]. Having a surface that avoids or retards the adhesion of ice, which is defined as icephobic, can reduce the power needed to protect the surfaces from ice accretion during the flight and revolutionize the technology implemented in large passengers’ aircraft (anti-icing) [13,14]. Novel light-weight ice protection system (IPS) with decreased power requirements can then be implemented in combination with advanced surfaces, replacing finally conventional IPS.

In the last decades, increasing interest in a novel technological concept, the hybrid ice protection systems (HIPS), was observed. HIPS includes an engineered surface combined with an ice protection system (IPS) [15]. Alternatively, the ice can be ultrasonically detached from the surface; however, ultrasonic initiation requires several hundred watts of energy and is, therefore, not ideal for the practical application [16,17]. In contrast to in-flight ice protection strategies, on ground ice accumulation is generally removed by ultrasonic methods, warmed up air, or by chemical coatings [15]. These coatings, which decrease the adhesion of water to the surface by reducing the free surface energy, are applied over large areas and are also not durable in the long term [18]. The mechanism behind this approach has shown that chemically-treated substrates, which are equipped with the superhydrophobic surface, have additional anti-icing properties. Furthermore, these chemically-treated surfaces have also shown a delayed frost formation for test conditions at temperatures of −6 °C. Despite the promising results, these solutions are not environmentally friendly, are time-consuming, and are related to high costs [19].

A novel approach to target these objectives are micro-textured functional surfaces with icephobic properties. An icephobic surface has per definition the following properties: (i) they delay the solidification time of the water or reduce ice accretion while the heterogeneous nucleation is hindered [20] and (ii) ice removal is promoted due to low adhesion strength between ice [21,22]. Superhydrophobic coatings on aluminum have demonstrated a remarkable potential to tackle the challenge of icing aircraft surfaces [23]. However, it should be mentioned that functional water-repellent surfaces are not always icephobic as well [24].

A fast and cost-efficient way to directly fabricate functional surfaces is laser surface texturing (LST). In LST, microstructures are generated on a surface by using laser-based methods like direct laser writing (DLW) or direct laser interference patterning (DLIP), among others [25]. In DLW, a single laser beam is guided by a scanner and focused by an optical element on the substrate. Depending on the laser wavelength, pulse duration, and pulse energy, a large variety of materials like metals, ceramics, and polymers can be processed [26,27]. For instance, this technology has been used to create superhydrophobic microstructures on aluminum as well as on titanium. Without applying chemical treatments, superhydrophobicity is reached with a static contact angle (SCA) above 150° and a sliding contact angle below 10° [28,29,30]. In addition to the topography, the chemical composition of the laser-generated texture also plays a role in the wettability modification. In fact, carbon-rich molecules can be absorbed from the ambient air, and the surface chemistry can be accordingly changed [31,32], making the wetting behavior time-dependent and its functionality limited for practical applications [33]. Moreover, microstructures fabricated on aluminum with surface features in the range of 30 to 50 µm have shown ice-delaying properties for single droplets freezing on the surface [34,35].

Another laser-based technology that can be applied for producing textured surfaces is direct laser interference patterning (DLIP). This method requires the use of at least two laser beams, which are superimposed on the substrate. This superimposition leads to an interference-generated modulation of the laser intensity distribution, which allows fabricating different periodic micro-patterns. The feature size of the pattern is usually in the range between ~500 nm and 10–20 µm [36,37]. In particular, the use of pulsed laser sources (nanosecond to femtosecond) has allowed the fabrication of features in the sub-µm range, even on metals [36,38]. Furthermore, DLIP has been widely employed for fabricating microstructures for biomedical applications on ceramics to influence the wettability on steel or to reduce the friction of automotive steer rings [39,40,41]. It is well known that hierarchical (also called multi-scale) microstructures are advantageous compared to a single-scale structure. For instance, they can provide multiple surface properties, including wear protection, ensuring a target functionality (e.g., antibacterial) for a longer time [42,43]. In addition, the superhydrophobic condition combined with an ice-repellent functionality has been also shown in recent work by creating 7.0 µm periodic pillars on aluminum as well as a hierarchical structure based on a combination of DLIP and DLW technology [35]. Nevertheless, ice repellency through microstructures has been only demonstrated under static testing conditions. In contrast, recently published results have shown that different laser-fabricated microstructures on titanium alloy may show dynamic ice-repellent properties, and a specific micro-texture design rule for minimizing the ice adhesion on titanium has been found [44]. A trend has been observed for surfaces in which a lower surface roughness R_z_ adds an advantageous effect on reducing ice adhesion [44]. By using a dynamic vibration testing technique with an electromagnetic oscillator, the ice adhesion and also its de-bonding forces can be determined accurately on a plethora of substrates [44,45]. However, the combination of different laser fabrication methods, the resulting hierarchical topographies, and their impact on ice adhesion, especially for aluminum, have not been investigated so far. Furthermore, to the best of our knowledge, it is not clear to what extent the roughness can still be reduced to further minimize ice adhesion.

In this study, various single- and multiscale patterns are fabricated on aluminum 2024 by means of DLW and DLIP. By combining both laser methods, complex hierarchical structures are introduced. The wetting characteristics are analyzed by static water contact angle and sliding angle measurements. The icephobic property is analyzed in relation to the adhesion of the formed ice on the laser-textured surfaces. Therefore, the terms “ice-repellent” and “icephobic” are to be understood in such a way that it requires reduced strength to release the ice from its surface. The performance regarding the dynamic ice adhesion is examined by interfacial shear stress experiments for rime, glaze, and mixed icing conditions using a lab-scaled icing wind tunnel. Hence, in these experiments, it is also feasible to take into account the wind speed and the droplet size, which allows a significantly more realistic view of the icing and deicing processes of laser-textured aluminum substrates.

## 2. Materials and Methods

### 2.1. Materials

The samples used in all experiments consisted of Al2024 (AMAG rolling GmbH, Ranshofen, Austria) in the shape of a cantilever with dimensions of 125 × 13 × 1 mm^3^. They were polished to a surface roughness R_z_ of 150 nm ± 40 nm. Prior to the laser processing, the samples were cleaned from contamination using isopropanol in an ultrasonic bath for 10 min. After the laser process, the structured samples were spray-coated with a hydrophobic agent (Mecasurf, perfluoropolyether compound by Surfactis, Angers, France) and stored under atmospheric conditions of 22 °C room temperature and 55% relative humidity. The non-flammable agent is composed of a chemically active perfluoropolyether compound dissolved in a fluorinated solvent, which is clear colorless and has an atmospheric lifetime of 0.77 years for a temperature range between −50 °C and +150 °C [46]. The used spray bottle (Buerkle GmbH, Bad Bellingen, Germany) was equipped with a 0.6 mm diameter nozzle, and the spray volume was 1.2 mL ± 0.1 mL per stroke. Five strokes were applied per sample with an interval of ~20 mm on the length-scale and a distance of ~100 mm to the surface in order to coat the cantilever homogeneously. This agent creates covalently bonded fluoride groups to cover the surface, saturating the material with non-polar compounds (reducing the surface energy) with an ultrathin layer (<5 nm). It is important to mention that the chemical reaction can take place only if the surface is not passivated with organic compounds: Therefore, the samples have been chemically treated directly after the laser process [41]. The compound reacts only with the surface of the material, while the unreacted compound evaporates due to the high volatility of the molecules, leaving a deposited monolayer [47].

### 2.2. Laser Structuring Methods

Two different structuring methods, direct laser writing (DLW) and direct laser interference patterning (DLIP), were used for fabricating the microstructures. For DLW processing, a fiber laser was used because the pulse duration could be adjusted and optimized for aluminum processing. For the DLIP experiments, however, a solid-state laser was used since it is especially characterized by its pulse stability for ultrashort laser pulses (ps range). Moreover, under determined texturing conditions, nanosecond laser pulses can create deeper micro-tranches on metal surfaces than picosecond ones due to the displacement of molten material [48]. The DLW technology is implemented by a laser surface texturing workstation (GF machining solutions P 600, Geneva, Switzerland). The system is featured with a galvanometer scanner system (Scanlab GmbH, Puchheim, Germany), including two mirrors and an IR (1064 nm) Ytterbium fiber laser (YLPN-1-4x200-30-M, IPG Laser GmbH, Burbach, Germany), providing a maximal output power of 30 W (Figure 1a). The pulse duration can be modified from 4 up to 200 ns, and the pulse repetition rate is flexible from 2 to 1000 kHz. In our experiments, 14 ns pulse durations were applied, and the repetition rate was set to 30 kHz. The beam was focused onto the sample using a 254 mm focal length F-theta objective, obtaining a spot with a diameter of 70 μm. The laser fluence was set to 1.06 J/cm^2^ by controlling the laser power. The laser beam was scanned along the surface of the sample with a constant speed of 250 mm/s. In order to achieve sufficient material removal, the structuring process was repeated 10 times in all cases. The mesh-like surface structures processed by this setup consisted of 3 sequentially formed line-like structures, whose orientation was each time rotated by 60°. The distance between the parallel lines was fixed to 50 μm, which is, therefore, defining the spatial period of the structures. Previous works using a similar texturing approach showed that this texture provided remarkable results in terms of water repellence and static icing properties and was, therefore, considered for the dynamic icing tests. A detailed description of the fabrication process was already published elsewhere [33,35,49].

For processing the samples with DLIP technology, a two-beam configuration was used. The used configuration utilizes an IR (1064 nm) solid-state Nd:YVO_4_ laser (Edgewave PX200, Würselen, Germany) with 10 ps pulses and maximum output power of 10 W. The workstation is equipped with a commercially available DLIP optical head (manufactured by Fraunhofer IWS, Dresden, Germany), which uses a diffractive optical element (DOE) to split the initial laser beam into two sub-beams (Figure 1b). These two beams are guided to a prism where they are parallelized and then focused to a beam radius ω < 100 µm on the sample surface using a 60 mm aspheric lens. Using this configuration, a line-like interference intensity profile is obtained within a circular area with a radius of ablated area rth of ~70 μm (Figure 1c). In this study, the intersection angle θ between the beams was set to 8.7° and 23.6° to obtain a spatial period Λ of 7.0 μm and 2.6° µm, respectively. The interference principle is depicted in Figure 1c, the laser fluence was set to 1.93 J/cm^2^, and the repetition rate of the laser was fixed to 10 kHz. The samples were translated in x- and y-directions by using high-precision axes (Aerotech, Pittsburgh, PA, USA) with an accuracy of ±2.5 μm. By moving the sample in the y-direction, the pulse-to-pulse overlap can be controlled, which finally determines the number of laser pulses, influencing the ablated structure depth. To obtain the pillar-like structures, it is necessary to rotate the samples by 90° and repeat the process.

The hierarchical structures, consisting of the DLW and DLIP, were manufactured by combining both strategies successively. A detailed description of the fabrication processes, the laser process windows, and the choice of the processing parameters was already published elsewhere [33,35,49].

### 2.3. Surface Characterization

To analyze the surface topography, scanning electron microscopy (ZEISS Supra 40VP, Jena, Germany) at an operating voltage of 15.0 kV was used. The wettability of the surfaces was characterized via static contact angle (SCA) and sliding angle (SA) measurements using the sessile drop method with a video-based optical goniometer (DSA 25, Krüss GmbH, Hamburg, Germany). This method was chosen due to its suitability for reproducible measuring. Deionized water droplets of 10 µL volume were deposited in atmospheric conditions for all measurements. The wetting tests were repeated 6 times to ensure statistical significance. Additionally, the real-to-projected area ratio Sdr was measured using a confocal microscope (Sensofar S Neox, Terrassa, Spain) with a 150× objective, resulting in vertical and lateral resolution of 2 nm and 140 nm. The surface roughness R_z_ was measured with a benchtop stylus profiler (DektakXT, Bruker, Billerica, MA, USA), following the standard DIN ISO 25178 and DIN EN ISO 4287, respectively [50,51]. For acquiring the results, a stylus with a 2 µm diameter was pressed on the surface with a weight of 2 mg and moved at a speed of approx. 0.05 mm/s. Every measurement point was repeated three times for a statistical purpose.

### 2.4. Ice Adhesion Test

To analyze the ice adhesion behavior for four different ice conditions, the interfacial shear stress was measured using a circular closed-loop wind tunnel iCORE (icing and Contamination Research facility, Airbus Defence and Space GmbH, Taufkirchen, Germany, Figure 2a) [50,52]. Therefore, the laser-structured area was mounted inside the test section, oriented against the flow direction, and the sample was fixed on one end to an electromagnetic shaker. Under previously defined conditions, ice accumulated on the structured area, which was then removed by inducing mechanical vibration (Figure 2b) [45]. The vibration caused shear stress at the interface between the accreted ice layer, and the cantilever was measured by a strain gauge and used to characterize the ice interfacial shear strength. The stress-strain curve, caused by the vibration, would be monitored during deformation, from which the failure stress could be identified microscopically [53]. The crack-propagation was afterward also visible (Figure 2c). In order to calculate the stress accurately, the ice thickness was measured and averaged at several points of the cantilever. Thus, the local waviness of the ice layer was taken into account. Furthermore, the dimensions of the cantilever were kept constant to ensure comparability of the results.

The calculation was performed under the assumption of a constant Young’s modulus of 9 GPa for each type of ice (E_ice_) based on several publications. It is worth mentioning that there were slight differences in the E_ice_ for the different icing conditions. Nevertheless, our assumption allowed comparability of the ice adhesion data within each ice type. For simulating atmospheric in-flight conditions, four different ice conditions (rime, mixed rime, glaze, and mixed glaze) were investigated by applying five testing parameters. These are the icing conditions to which an aircraft mostly can be exposed during a flight. They result from the varying flight altitude, which is associated with different air temperatures and ice-water mixtures. For instance, the glaze ice condition usually occurs at an altitude around 2000 m at a temperature range between 0 °C and −10 °C. Contrarily, the rime ice condition appears at altitudes above 3000 m and at a temperature of at least −20 °C. Furthermore, different ice conditions are characterized by varying densities and hardnesses [15].

The ice-producing parameters were related to the icing design envelope specifications from the Federal Aviation Administration (FAA) [54]. The influencing parameters were beside the total air temperature (−5 °C and −20 °C) and the airspeed (50 m/s and 80 m/s), the liquid water content (LWC), the mean effective volume droplet diameter (MVD), and the approximate freezing fraction (AFF). The MVD is defined as the midpoint (median) droplet size, 20 µm in our case, where half of the water volume in the cloud is in droplets smaller, and half of the volume is in droplets larger than the median [54]. The approximate freezing fraction (AFF) determines the fraction of water that freezes initially when water droplets impinge on a surface. For instance, for rime ice, the freezing fraction is 1, which means all the impinging water is freezing immediately. In contrast to that, for the glaze, the freezing fraction is close to 0, and most of the impinging water will stay liquid and run back [55]. The LWC defines the density of super-cooled water droplets in the icing cloud and is ranging between 0.3 and 1. An overview of the parameters for creating the icing regimes is given in Table A1 in the Appendix A.

## 3. Results and Discussion

### 3.1. Laser Structuring of Aluminum Substrates

For the structuring process of the aluminum substrates, DLW and DLIP methods were used to create different topographies, with feature sizes ranging from 2.6 µm to 50 µm, employing ps and ns laser sources, respectively. In contrast to the untextured Al reference (Figure 3a), the DLW-treated substrates showed a morphology dominated by melting processes due to the used nanosecond pulse duration. In particular, the mesh-like structure with a repetitive feature size of 50 µm consisted of molten and re-solidified material, as shown in Figure 3b. Furthermore, the topography of the DLW-treated samples was assessed through confocal imaging, revealing a roughness R_z_ of ~43.1 µm and an average structure depth of 35.7 µm. Significantly less melting occurred on the DLIP-treated samples due to the ps-based micro-processing. Applying a spatial period of 7.0 µm, perfectly ordered pillar-like microstructures were fabricated, and a roughness R_z_ of ~4.1 µm was reached (Figure 3c). To produce a multiscale pattern, the ns-DLW and ps-DLIP technologies were combined, resulting in a hierarchical structure consisting of 7.0 µm pillars on the 50 µm meshes. The ps-pulsed laser source was selected in order to avoid further thermal treatments of the DLW structures. The added DLIP-pillars led to a slight decrease of the roughness (compared to DLW) to an R_z_ of ~38.1 µm (Figure 3d). It could be assumed that the additional DLIP process slightly smoothened the roughness peaks caused by the ns-DLW melting since the utilization of the ps laser sources resulted in ablation and vaporization of the treated material.

In addition to the above-mentioned patterns, also line-like and cross-like structures with lower spatial periods were produced. These had a spatial period of 2.6 µm and a surface roughness R_z_ of 0.61 µm for a line-like and 0.96 µm for a cross-like pattern (Figure 3e,f), showing a significant lower roughness compared to the 7.0 µm pillar-like structures.

The ice adhesion functionality of a surface is dominated by the surface’s topography and chemistry, as mentioned in the introduction. It is well-known that laser processing of aluminum surfaces leads to an influence of the chemical composition on the textured surface [33,44]. Directly after laser processing, the structured substrates and the untextured reference were spray-coated using a perfluoropolyether (PFPE) monolayer coating, which reduces the surface tension and ensures constant chemical surface conditions all over the surface. PFPE is very inert, and the chemical bond has high stability due to the covalent carbon-fluorine bond [41,50,56]. After the treatment, all samples showed a superhydrophobic characteristic due to the lower surface tension combined with the produced micro-texture, regardless of the size or orientation of the microstructures. In particular, the DLIP-produced cross-like patterns showed static water contact angles (SWCA) of 164°, and the hierarchical pattern an angle of 172°. The untreated reference remained in the hydrophilic condition with an SWCA of 59°, and the PFPE coated reference led to a hydrophobic transition with an SWCA of 122°. All angles are reported in Table 1. The obtained results clearly show the influence of the surface pattern on the wetting properties of the samples and confirms that a spray-coated and untextured surface does not develop a superhydrophobic characteristic [41,44].

While the static water contact angle is commonly known as a criterion to characterize the wettability of a surface, it does not necessarily allow to determine the (super-)hydrophobic character of a surface. Therefore, the sliding angle (SA), denoted as the tilting angle where the droplet starts to slide on the surface, is an additional parameter that has to be considered [57]. In this study, all laser-fabricated structures exhibited a sliding angle below 10°, allowing the samples to be considered as fully superhydrophobic (see Table 1) [58,59]. In detail, the sliding angles for the microstructured surfaces ranged between 2° (DLW and DLW + DLIP) and 9° (DLIP cross). Both references did not show any sliding characteristics due to their hydrophilic and moderate hydrophobic surface state. The geometries and their periodicities, as well as the wetting characteristics (SCA and SA) and the roughness R_z_ of the considered topographies, are summarized in Table 1. A detailed description of the data acquisition is given in the experimental section.

It is important to state that the determination of the wetting contact angle is representative of the macroscopic wetting behavior of the treated surfaces, and it is not sufficient to describe the wetting phenomena involved during in-flight (icing) conditions [20]. In fact, as stated in the introductory part, supercooled microscopic droplet imping at high velocity on metallic surfaces, making the effect of surface tension, hydrostatic pressure, and surface microstructure size more relevant [4,60].

It is worth mentioning that the durability of the laser-generated structures and their chemical composition when exposed to consecutive icing/de-icing cycles was examined in a previous study [50]. It was demonstrated that the chemical functionalization with the PFPE substrate after laser processing produced surfaces that preserved their superhydrophobicity beyond 16 icing/de-icing cycles, indicating a likelihood of covalent bonds between perfluoropolyether and metal oxide. The same observation was found in the samples presented in this study. The SCA and SA did not decrease after consecutive icing/de-icing cycles.

### 3.2. Ice Adhesion Analyzed by Mechanically Induced Stress

It is well-known that superhydrophobic aluminum leads to delayed ice-formation due to the reduced contact area between a single water droplet and the surface [35]. However, most of the works present in literature are based on the icing of water droplets under static conditions. An alternative to the static analysis of the anti-icing properties are methods in which the surface is mechanically strained while measuring the ice adhesion [61]. It is fundamental that the surface roughness and the microstructure are optimized in order to allow proper anti-icing functionality [44,62]. In fact, it has been demonstrated that water-repellent surfaces with a spatial period of 35 µm even lead to a stronger ice adhesion compared to an untreated titanium reference [50].

The microstructures in this study were fabricated on 125 × 13 × 1 mm^3^ cantilevers, which were placed afterward in an icing wind tunnel in order to investigate the ice adhesion. Four icing conditions were analyzed, which simulated flight conditions encountered in different atmospheric regimes. These generated four qualities of ice were rime, mixed rime, mixed glaze, and glaze ice. After that ice was accreted on the surface of the cantilever, the icing cloud was turned off, and an oscillation was induced on the cantilever through a piezoelectric transducer. A linear frequency sweep was applied, letting this increase with time and causing an increase in the amplitude of the oscillation when the resonance frequency of the oscillating body was reached [45]. Since the resonance frequency depends on the thickness and density of the accreted ice, this differed among the investigated samples (60–70 Hz). Once a critical amplitude value was reached, the ice detached from the surface (crack initiation), and the interfacial shear stress could be retrieved and quantified. A strain gauge was placed on the backside of the cantilever in order to detect the strain during the oscillation and display the change of the mechanical properties of the composite beam (i.e., ice and metal) due to ice detachment. An example of a strain gauge reading is shown in Figure 4 for the untextured reference, DLW, and DLW + DLIP treated surfaces. In all cases, the amplitude changed drastically at 0.75 s. Such variation was related to a discontinuity in the mechanical properties of the composite beam due to an interfacial crack between ice and cantilever.

The amplitude of the strain recorded right before the interfacial crack represented the limit at which the interfacial shear stress matched the adhesion strength of the ice to the surfaces. Exceeding this value triggered a crack at the ice-surface interface. The Euler-Bernoulli beam theory could be used to correlate the strain to the maximum interfacial shear stress (i.e., ice adhesion) between ice and surface, which is expressed in Equation (1) [45]:(1)$$\tau_{int}=\frac{\varepsilon_{EF-al}E_{ice}\left(h_{ice}^{2}+2h_{ice}\left|e\right|\right)}{2\left(x-l\right)\left(h_{cl}-\left|e\right|\right)},$$
where $\tau_{int}$ is the interfacial shear stress, $\varepsilon_{EF-al}$ is the amplitude of the strain measured by the strain gauge, $E_{ice}$ is the Young’s Modulus of ice, $h_{ice}$ is the thickness of the ice, $h_{cl}$ is the thickness of the cantilever, x is the distance of the strain gauge to the fixed end of the cantilever, l is the length of the cantilever, and e is the eccentricity of the neutral axis of the ice/metal beam with respect to the ice/metal interface [53]. The eccentricity of the neutral axis could be calculated from Equation (2):(2)$$e=\frac{\left(h_{cl}^{2}+\frac{E_{ice}}{E_{cl}}h_{ice}^{2}\right)}{2\left(h_{cl}+\frac{E_{ice}}{E_{cl}}h_{ice}\right)},$$
where $E_{cl}$ is the Young’s Modulus of the bulk cantilever material. The Young’s Modulus of ice used for shear adhesion strength calculation was taken from prior research for fully dense ice [63,64]. As a result, a low shear stress value resulted in less mechanical force necessary to remove the ice from the surface [65]. In general, the lower the maximum amplitude of the strain before the interfacial crack is, the lower the ice adhesion (with all the other parameters fixed). Therefore, based on the three shown exemplary measurements from Figure 4, it can be concluded that this method is suitable to quantitatively evaluate the adhesion of ice under different conditions [45]. It is worth to mention that other authors have also reported superhydrophobic aluminum surfaces tested in an ice wind tunnel, leading to a significant delay of the initiation of ice formation [66]. Moreover, it was demonstrated that the ice adhesion was reduced from ~360 kPa to ~80 kPa for coated and etched superhydrophobic aluminum (AA6061-T6) substrates, which corresponded to a reduction of approximately 78%. The samples were tested exclusively for the glaze ice condition in a centrifuge set-up [67].

In contrast to that, the results of the ice adhesion tests for the seven samples (two references and five micro-structured cantilevers) by means of the retrieved interfacial shear stress under four different icing conditions are summarized in Figure 5.

Each measurement was repeated 4 times for each surface type and for each ice condition (in total, 96 measurements). The average value of one measurement series defines the height of the bar, while the error bars result from the standard deviation of each measurement series. The term reference refers to the untextured sample, and it provides average interfacial shear stress of around 36 kPa, whereas, for mixed rime, mixed glaze, and glaze conditions, the stress was slightly increased to around 50 kPa. Besides that, some trends were observed, which are discussed in the following.

The DLW structure (50 µm), the pillar-like DLIP structure (7 µm), and the hierarchical pattern showed interfacial shear stress between 48 and 64 kPa, which was similar (for mixed glaze ice) or even significantly higher (up to 107 kPa for rime and mixed rime ice) compared to the reference. It could be assumed that the roughness R_z_ of 43.15 µm, 4.16 µm, and 38.16 µm (DLW, DLIP, and DLW+DLIP) resulting from the coarse feature sizes of 50 µm (DLW) or 7 µm (DLIP) provided a favorable topography for the droplets to attach and interlock. Note that the mean effective volume droplet diameter (MVD) of the impinging droplets was ~20 µm and thus in the order of magnitude of the investigated microstructures (see Table A1 in the Appendix A).

In contrast, the DLIP structures with a spatial period of 2.6 µm showed a significantly lower interfacial shear stress. The stress for the smallest examined texture was between 27 and 38 kPa (DLIP line-like 2.6 µm) and even lower between 6 and 10 kPa (DLIP cross-like 2.6 µm) for the icing condition of rime, mixed rime, and mixed glaze. This amounted to a shear stress reduction of ~90% compared to the reference. This behavior could be explained with a synergic effect of high hydrophobicity and low surface roughness.

Besides that, it is worth mentioning that both references did not show superhydrophobic characteristics for all icing conditions. This property is fundamentally necessary since impacting droplets will not adhere to the surface. The droplet rolls off the surface because the anti-wetting capillary pressure is greater than the dynamic and water hammer wetting pressures [68].

In fact, the texture size produced on the “DLIP cross” sample is low enough to both repel micrometer-sized water droplets and to prevent the formation of large anchor points for the ice adhesion and accretion [69]. Although having similar roughness, the “DLIP line” sample shows instead a much higher ice adhesion. This can be explained by the surface modifications in the micro-scale, which only take place in one direction (line-like texture). Hence, the overall response to ice adhesion is, therefore, higher than for an isotropic texture (cross-like). Therefore, it can be assumed that a line-like geometry provides a preferred orientation regarding ice accumulation. Such assumption finds support in the work from Lo et al. [70]. Their research was focused on ice nucleation studies of different surfaces using environmental SEM (ESEM). The authors showed that ice nucleation was controlled and confined on V-shaped microgroove patterned surfaces. The ice is showing preferential nucleation sites along the manufactured grooves. This anchoring theory is also valid for the hierarchical DLW+DLIP structure with a lower roughness (38.16 μm) compared to the DLW structures (43.15 μm). However, this topography seems to offer more anchoring possibilities for ice formation. Due to the hierarchical surface, this hypothesis is especially claimed for lower temperatures, which are present in the rime and mixed rime conditions at −20 °C.

### 3.3. Influence of the Surface Topography on the Ice Adhesion

To further investigate the influence of the surface peak-to-peak-roughness, the interfacial shear stress was plotted as a logarithmic function of the real-to-projected area ratio Sdr in %, for the four ice adhesion characteristics (Figure 6). The corresponding roughness and Sdr parameters are additionally given in Table 1.

Although the experimental data did not show a clear trend, it was evident that a defined Sdr ratio was favorable in order to reach low interfacial shear stress under rime, mixed rime, and mixed glaze conditions. The differences of the Sdr values for the 50 µm DLW and 2.6 µm DLIP textures could be ascribed in-depth and the aspect ratios of the textures. Since the 2.6 µm textures were a few hundred nm deep, the aspect ratio was consequently low and in the range of ~0.1. In contrast, the DLW textures showed a structure depth of ~35.7 µm, which resulted then in an aspect ratio of ~0.7. As a result, the DLW textures provided a significantly higher Sdr ratio. The Sdr ratio was in the range of ~13 (for the DLIP cross 2.6 µm textures), corresponding to a surface roughness R_z_ of 0.96 µm, and resulted in the detachment of the ice from the aluminum surface at interfacial shear stress below 10 kPa. However, a direct correlation of the individual textures regarding their ice adhesion, as shown in Figure 6, was difficult due to their significantly diverse topography. Based on the experimental results, it could be concluded that a smoother surface seemed to promote a higher adhesion [71].

These findings were supported by previous studies reporting the systematic reduction of ice adhesion strength with the decrease of surface roughness regardless of surface wettability [72,73]. In-situ icing studies indicated that ice did not anchor on smoothened metallic surfaces but anchored on the as-received metallic surfaces, which is consistent with the results in Figure 6 [73]. The larger microstructures (DLIP 7.0 µm, DLW 50 µm, and DLW + DLIP) showed significantly higher ice adhesion stress, which was related to the average effective drop diameter (MVD). When the 20 µm droplets hit the textured surface, especially the deep DLIP pillars or the DLW structures (Sdr between 130 and 240%) offered space between the microscopic features. The ice adhesion was, in this case, even significantly higher compared to the reference and the DLIP structures with the 2.6 µm structures. This hypothesis was further supported by earlier research results obtained on titanium substrates [44].

Apart from the roughness and the Sdr parameter, it was observed that not only the spatial period influenced the ice adhesion but also the pattern geometry. Two samples were fabricated with the same spatial period of 2.6 µm with line and cross-like geometries (see Figure 3e,f). Nevertheless, the cross-like geometry showed substantial lower ice adhesion. It was further demonstrated that the icing condition also had a significant influence on the shear stress. For example, it has been shown that for the glaze condition, increased stress was measured for all microstructured surfaces than for the reference (see Figure 6, glaze). This could be linked to the combination of a higher temperature (−5 °C), higher speed (80 m/s), and high liquid water content (LWC) of 1 g/m^3^ (see Table A1 in the Appendix A). Under this condition, the pressure generated at the impact could cause the partial wetting of the structures, given the nature of glaze ice [68]. As a matter of fact, only a lower portion of water is freezing on impact, and the rest is able to run wet and freeze afterward. As a consequence, liquid water can penetrate within the laser-generated structures and freeze in them, increasing the mechanical interlocking between ice and surface. Subsequently, it was reported in other studies that the gravity effect on the droplet trajectories favored ice formation in this case [60,74].

In summary, the DLIP 2.6 µm line and cross are the only laser-treated surfaces, exhibiting remarkable icephobic properties in this study. The cross-like strategy is performing better, most likely due to its isotropic morphology, which is reducing the amount of anchoring points for the ice formation. The DLIP 7.0 µm and the DLW 50 µm are not able to repel supercooled water droplets, despite their superhydrophobicity [44]. However, such surfaces could still be used for repelling larger droplets, possibly at lower impact speeds. The combination of DLW and DLIP does not provide any benefit in terms of ice adhesion reduction in dynamic conditions, although previous results remarked the advantages when combining both geometries in static conditions [35,75,76].

## 4. Conclusions

In this work, five different periodic patterns were textured on AA2024 clad cantilever samples using direct laser interference patterning, direct laser writing, and a combination of both methods. Additionally, two untextured reference samples were considered. The spatial period of the textures ranged from 2.6 µm to 50 µm, showed a line-, cross-, pillar-, or mesh-like pattern, and covered a corresponding roughness R_z_ between 0.61 µm and 43.15 µm linked to an Sdr ratio between 5.9 and 240. After a chemical post-treatment, all textures showed superhydrophobic wetting characteristics with a static contact angle over 166° and a sliding angle below 10°. This combination of material processing, laser texturing, and the subsequent chemical treatment was conducted for the first time on aluminum cantilevers. Using a dynamic vibration testing technique, the shear stress between the structured sample and ice was investigated. Each structural geometry was tested under four different icing conditions, namely rime ice, glaze ice, and two mixed ice conditions. For the rime ice and mixed ice conditions, the cross-like DLIP pattern with a size of 2.6 µm showed the lowest interfacial shear stress, between 6 and 10 kPa. In contrast, the stress of the reference was between 36 and 57 kPa independently of the icing condition. This magnitude of shear stress is extraordinarily low and has not been demonstrated by any known alternative approach under these icing conditions to date to the best of our knowledge. Surface patterns with larger feature sizes, such as 7.0 µm DLIP, 50 µm DLW, or the hierarchical structure, showed even higher shear stresses (between 80 and 108 kPa) compared to the reference. Consequently, it can be stated that microstructures with feature sizes smaller than 5 µm and a particular structure type (line-like and cross-like) are especially suitable for designing icephobic surfaces. Using the results of these experiments, the field of the perfect icephobic texture for real dynamic icing conditions can be further narrowed down, and future research can build on this knowledge. The results obtained and the methodology developed in this work can be further adapted to other alloys and even polymers, unlocking completely new application examples in the challenge facing icing surfaces in the fields of aerospace technology.

## Figures and Tables

**Figure 1 nanomaterials-11-00135-f001:**
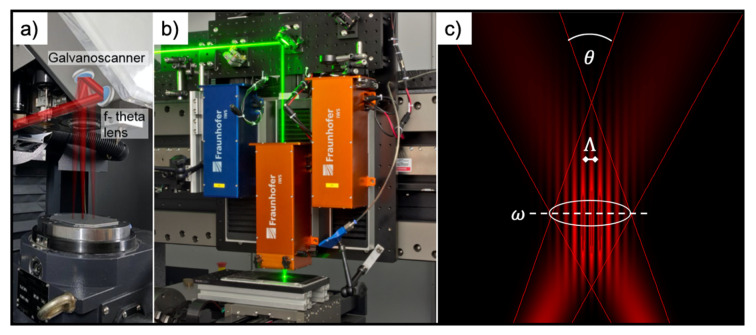
DLW (direct laser writing) configuration equipped with a galvanoscanner and an f-theta-lens used in the experiments (**a**), 3D-DLIP workstation (developed at the Technische Universität Dresden) with a 1064 nm wavelength laser source emitting 10 ps pulses for the processing of 2D and 3D parts ((**b**), the visible laser beam illustrates the process), graphical representation of the used two-beam direct laser interference patterning (DLIP) principle, showing the most relevant parameters describing both the pattern and the process: ω—beam diameter; θ—beams intersection angle; Λ—laser intensity profile/ablated structure period (**c**) [35,42].

**Figure 2 nanomaterials-11-00135-f002:**
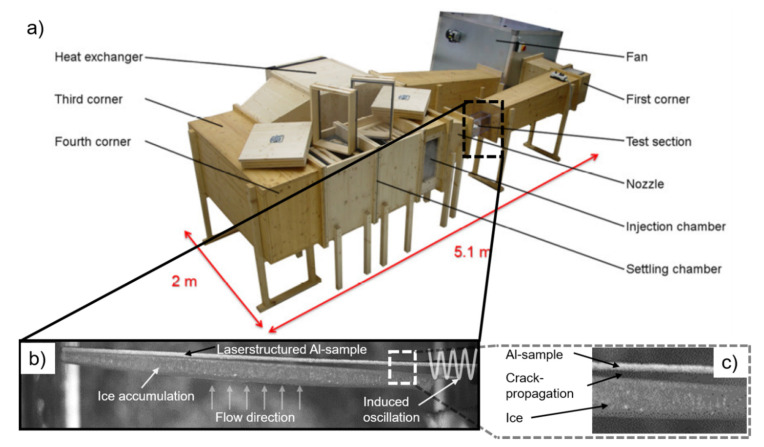
Icing wind tunnel iCORE (icing and Contamination Research facility), which allows adjusting the wind speed and temperature by accessing a fan and a heat exchanger. The droplet size and stream water content in the test section are set by the settling and injection chambers before the nozzle (**a**) [51]. In the test section, ice accumulates on the laser-structured sample, and the specimen is exposed to mechanical shear stress due to induced oscillation (**b**), which finally leads to crack propagation (**c**).

**Figure 3 nanomaterials-11-00135-f003:**
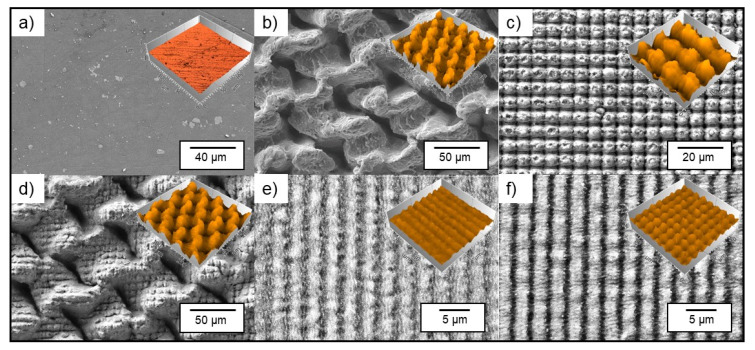
Scanning electron microscope images of (**a**) an untextured reference and of microstructures fabricated with (**b**) DLW for a 50 µm mesh-like structure using a laser fluence of 1.06 J/cm^2^, a scanning speed of 250 mm/s at a repetition rate of 30 kHz; (**c**) DLIP for a 7.0 µm pillar-like structure using a laser fluence of 1.93 J/cm^2^, a pulse-to-pulse overlap of 99% at a repetition rate of 10 kHz; (**d**) a combination of DLW and 7.0 µm DLIP for a multiscale structure; (**e**) DLIP for a 2.6 µm line-like structure and (**f**) DLIP for 2.6 µm cross-like structure using the same fluence, repetition rate, and overlap as for the 7.0 µm DLIP structures. The inserted confocal images provide an impression of the topography. The laser parameters are provided in the experimental section.

**Figure 4 nanomaterials-11-00135-f004:**
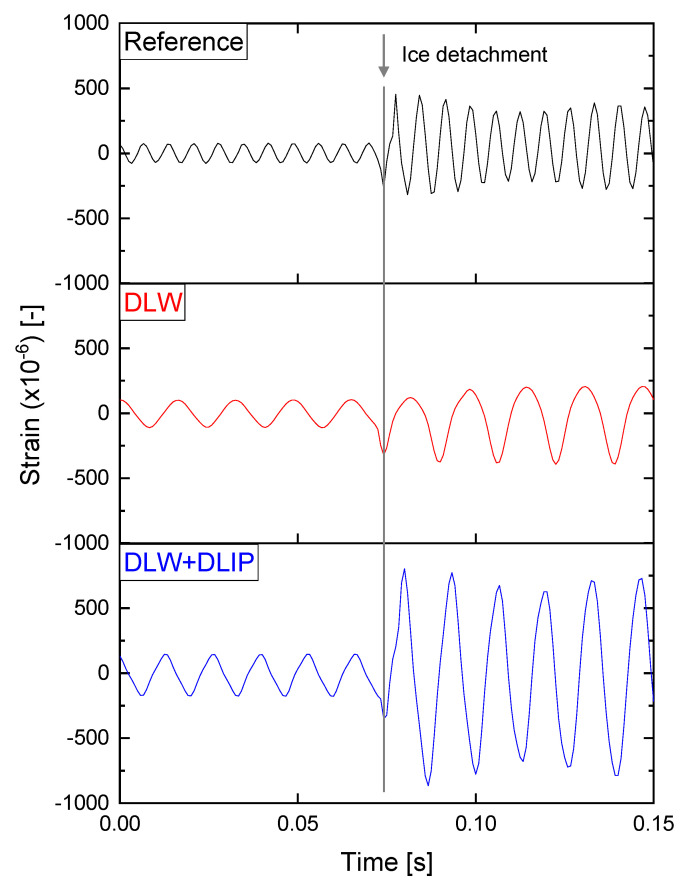
Example of strain gauge readings recorded during the ice adhesion tests for three different samples of an untextured reference AA2024 (**top**), DLW (**middle**), and DLW + DLIP (**bottom**) under test conditions of mixed/rime ice. The time range in which the ice detachment occurred was framed to 0.15 s to obtain a more detailed overview.

**Figure 5 nanomaterials-11-00135-f005:**
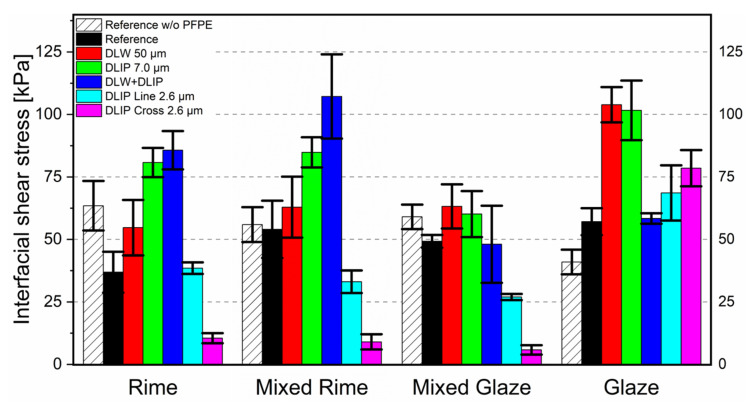
Histograms depicting the results of the ice adhesion tests for the four different ice conditions of rime, mixed rime, mixed glaze, and glaze with the interfacial shear stress for the five laser-structured specimens, a reference, and a PFPE untreated (w/o PFPE) reference.

**Figure 6 nanomaterials-11-00135-f006:**
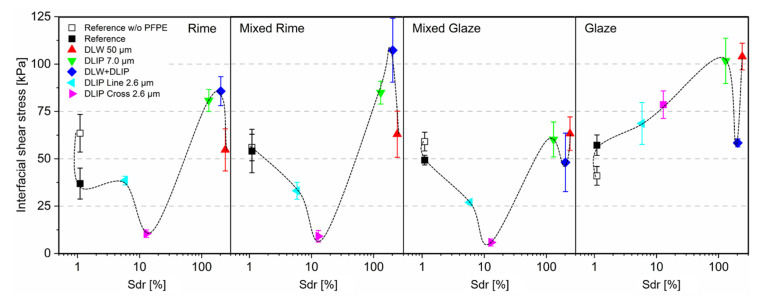
Interfacial shear stress of the laser-treated samples and the two references as a logarithmic function of the Sdr ratio for the icing conditions of rime, mixed rime, mixed glaze, and glaze. The black dashed spline is meant only to guide the eye of the reader.

**Table 1 nanomaterials-11-00135-t001:** Topographical and wetting characteristics of the fabricated samples treated with perfluoropolyether (PFPE), direct laser writing (DLW), direct laser interference patterning (DLIP) and an combination of DLW and DLIP (DLW/DLIP).

Parameter	Untreated Reference	Reference with PFPE	DLW 50 µm	DLIP 7.0 µm	DLW/DLIP 50/7.0 µm	DLIP Line 2.6 µm	DLIP Cross 2.6 µm
Geometry	-	-	Mesh-like	Pillar-like	Hierarchical	Line-like	Cross-like
Spatial period (µm)	-	-	50	7	50/7	2.6	2.6
Static contact angle (°)	59 ± 2	122 ± 2	171 ± 2	165 ± 1	172 ± 1	166 ± 1	164 ± 3
Sliding angle (°)	No sliding	No sliding	2 ± 1	4 ± 2	2 ± 1	3 ± 1	9 ± 4
Roughness R_z_ (µm)	0.15 ± 0.04	0.18 ± 0.04	43.15 ± 0.32	4.16 ± 0.39	38.16 ± 2.33	0.61 ± 0.03	0.96 ± 0.17
Real area to projected area ratio Sdr (%)	1.1	1.1	240	130	203	5.9	12.9

## Data Availability

Data is contained within the article.

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
