# Peer review of "Icephobic Performance of Multi-Scale Laser-Textured Aluminum Surfaces for Aeronautic Applications"

_nanomaterials, 2021, doi:10.3390/nano11010135_

Round 1
Reviewer 1 Report
The effects of superhydrophobicity and morphology of textured surfaces on icephobicity and deicing have profound technological implications and their interrelations are still unclear. The data presented here can provide some insights to resolving these issues. This paper presents the icephobic performance of different textured surfaces coated with a monolayer of PFPE. Different textured surfaces were fabricated by using ns and ps pulsed laser systems which provide the ability to use Direct Laser Writing (DLW), Direct Laser Interference Patterning (DLIP) and a combination of both techniques. The PFPE coated texture surfaces have excellent superhydrobic characteristics based on the static contact angles (SCA) and sliding angles (SA). The reported icephobicity based on experimentally determined interfacial bonding strength showed the textured patterns with the smallest featured size studied viz. 2.6 um exhibit the lowest interfacial shear stress thereby suggested that textured surfaces with sizes <5um are suitable for designing icephobic surfaces. Here are some comments:
- Table 2: Reference with PEPE SA > 90 degree is too large.
- Table 2: DLIP cross 2.6um, SA of 9 degree is very high compared to DLIP line 2.6um.
- Need to explain why Sdr larger for 50um compared to 2.6um, normally the reverse is expected.
- Is stylus profiler the standard tool to measure Sdr.
- Do SCA and SA changed after the experiments due to the delamination of the PFPE coating which is bonded to the aluminum surface by weak van der Waals interactions.
- How interfacial shear stress correlate with icephobocity.
Reviewer 2 Report
In this manuscript the authors has described the research of the multi-scale laser-textured aluminum surfaces for aeronautic applications.
In my opinion there are several assumptions employed in this work, that should be examined.
Comments and Suggestions for Authors
- The abstract needs to include the main impact of this study. The abstract should include some short quantitative conclusions.
- There is no comparison of the results with those of other authors.
- Table 1 and 2 are too large.
- A list of symbols should be added to this manuscript.
- The conclusion section should be extended. It should include the novelty of this research and important results obtained from this work.
Reviewer 3 Report
The manuscript describes the results of experimental study into the effect of surface laser texturing on ice adhesion on aluminum. Certain improvement was obtained by the combination of two texturiung technologies. The study is well organized. The results are sound,. The conclusions are supported by the experimental evidences.
The manuscript is too lengthy. The authors should centre on the main objectives and move various secondary aspects (type of ice formation, etc.) to supplementary materials. There are several redundancies, which can be avoided. Some minor grammar corrections are neccessary.
Reviewer 4 Report
This presented work proposes an approach (particularly a combination of DLW and DLIP) to improve the icephobic property of Al surface, which would be of great importance for aeronautic applications. Although the technical aspect of this work is sound and fairly well presented, I found this submitted manuscript to be a slight improvement of a previously published report in Scientific reports (Scientific Reports volume 9, Article number: 13944 (2019)). The slight mentioned improvement includes an ice adhesion test using a icing wind tunnel.
The authors should therefore emphasize the scientific value/advancement of this work compared to their report in Sci. Rep. in 2019.
Round 2
Reviewer 4 Report
the authors have addressed my comment.